# High dropout rate from maternity continuum of care after antenatal care booking and its associated factors among reproductive age women in Ethiopia, Evidence from Demographic and Health Survey 2016

**Atalay Goshu Muluneh**[1]*, **Getahun Molla Kassa**[1], **Geta Asrade Alemayehu**[2], **Mehari Woldemariam Merid**[1]

**1** Department of Epidemiology and Biostatistics, Institute of Public Health, College of Medicine and Health Sciences, University of Gondar, Gondar, Ethiopia, **2** Department of Health System and Policy, Institute of Public Health, College of Medicine and Health Sciences, University of Gondar, Gondar, Ethiopia

* goshuatalay12@gmail.com, atalayg1921@gmail.com

**Data Availability Statement:** The data used for the preparation of this manuscript were taken from the

## Abstract

### Background

Maternal continuums of care were vital to reducing maternal and neonatal mortalities. While the dropout rate remains high and limited studies were found on risk factors associated with a high dropout rate of the maternal continuum of care.

### Objective

This study aimed to assess the magnitude of dropout rate and its associated factors of maternity continuum of care in Ethiopia, 2016

### Methods

An in-depth secondary data analysis was conducted from the Ethiopian Demographic and Health Survey 2016 data. A total of 4,693 women who were booked for antenatal care visit were included to the final analysis. A community-based cross-sectional study design and a pre-tested and standardized questionnaire were used to collect the survey data. Data were weighted using women data weighting variables. Chi-square and multicollinearity assumptions were checked for independent variables. Bi-variable and multivariable logistics regression used to identify associated factors with a cut of the p-value of 0.2 and 0.05 respectively. Adjusted Odds Ratio (AOR) with 95%CI was reported for the final model.

### Results

Of the total 4,693 women who were booked for antenatal care visits, 2,092(44.58%), 2,183 (46.52%), and 4,086(87.07%) dropped from a recommended number of ANC, Institutional delivery and postnatal care visit respectively. Only 308 (6.56%, 95%CI: 5.89, 7.31) women

"Ethiopian Demographic and Health Survey 2016". Users can access the data at http://www.dhsprogram.com.

**Funding:** No funding organization.

**Competing interests:** The authors declare that they have no competing interests.

**Abbreviations:** ANC, Ante Natal Care; AOR, Adjusted Odds Ratio; EDHS, Ethiopian Demographic and Health Survey; EA, Enumeration Area; PNC, Postnatal Care.

used all the complete continuum of care. Not married, and poorest wealth index were significantly associated with dropout from ANC visit. Being a protestant religious follower was significantly associated with dropout from PNC after antenatal care booking. While not exposed to media, distance from health facility as a big problem, protestant affiliation, parity of 2 to 4 and above4, Wealth index of the poorest, poorer, middle, and richer significantly associated with dropout from institutional delivery. Not being informed about pregnancy complications during their ANC visit was significantly associated with dropout from ANC, PNC, and institutional delivery.

## Conclusions

Dropout of women from the maternity continuum of care after antenatal care booking was a public health problem in Ethiopia. Socio-demographic, pregnancy, and health service-related factors were significant determinants of dropout from the maternity continuum of care. Improving the family wealth index, increasing access to health facilities, media exposure, and giving more information during the antenatal care visit is important to reduce the dropout rate from the maternity continuum of care.

## Background

Maternal mortality remains unacceptably high and Sub-Saharan Africa alone accounts a 2/3$^{rd}$ of global maternal mortality [1]. Maternal and child health were a great concern for the Ethiopian ministry of health [2]. Ethiopian women had a 21 per 1,000 women lifetime risk of death related to pregnancy and a maternal mortality ratio of 412 per 100,000 live births[3]. Antenatal care visits, health facility delivery services are very important to reduce maternal and neonatal morbidities and mortalities [4]. The maternity continuum of care was very vital to reduce maternal and child mortality to achieve the sustainable development goal [5, 6]. Antenatal care services were started across the globe to reduce maternal and neonatal mortality by increasing skilled birth attendance and institutional delivery rate [7–10]. The World Health Organization (WHO) and other stakeholders are fighting to reduce maternal and child mortality with different intervention programs and strategies [11]. Ethiopia strives to end maternal and neonatal mortalities through increased production of skilled professionals on maternal and child health, collaboration with different governmental and non-governmental organizations, increased budget allocation, and give special emphasis [2, 4]. In Ethiopia, 43% of women used four and above ANC, 48% had institutional delivery and 34% of women had PNC visits [4]. Although the country has made remarkable achievements in reducing maternal and child morbidity and mortality, neonatal and maternal health has remained a public health problem [4]. The increasing dropout rates from each maternal continuum of care takes the lion share for high maternal and neonatal mortality in the country [3, 4].

Women in developing countries including Ethiopia were vulnerable to high dropout rates from the maternal continuum of care sequentially from Antenatal Care (ANC) to institutional delivery, and from institutional delivery to Postnatal Care (PNC) services [12–17].

Poverty, distance from the health facility, lack of information, inadequate and poor quality services, cultural beliefs, and practices were the main factors that affect women in developing countries to receive care during pregnancy and childbirth [1, 18]. Different factors also contribute to completion/dropout rate of the maternal continuum of care age at first birth, the

number of children, with higher education, belonging to richest quintile, place of residence, women autonomy, and mass media exposure [12, 15, 16, 19].

This study was conducted to determine the dropout rate and associated risk factors of the maternity continuum of care in Ethiopia based on the national demographic and health survey 2016 data. The findings of this study will provide important input for policymakers about the continuity and possible factors of the maternal continuum of care services in Ethiopia.

## Methods

### Study design and period

In-depth Secondary data analysis was employed from the Ethiopian Demographic and Health Survey (EDHS) 2016 data. The survey data was a nationwide cross-sectional data that has been collected in Ethiopia and we use a case-control study design for our analysis. The detail of the methodology and study design could be found in the EDHS 2016 report [3].

### Study area

Ethiopia is the 2nd most populous country in Africa with a high fertility rate, maternal, and neonatal mortality. Low utilization of maternity continuum of care services was a major problem of the nation and common contributors to maternal and neonatal mortality [3]. The country had nine regions divided into Zones and each zone was divided into Woredas. Finally, Woredas are divided into smallest administrative Units called Kebeles. In Ethiopia, as in most African countries, women play the principal roles in the rearing of children and the management of family affairs. Ethiopian Ministry of ealth gives maternal health services like Delivery, Antenatal care, and postnatal care services free of charge in all public health facilities.

### Data source and measurements

Every five years, the Demographic and Health Survey of Ethiopia (EDHS) collects data at the national level based on representative samples and key indicators including maternal health conditions. Interviewer administered questionnaire was used to collect data on women of reproductive age (15–49) years. The questionnaire includes socio-demographic, socio-economic pregnancy, and maternal health service-related variables related to women's health. A stratified two-stage cluster sampling; with 645 Enumeration Areas(EAs) (202 in Urban and 443 in rural areas) were selected with probability proportional to EA size. A total of 15,683 women were interview for maternal health and 7,589 women who give birth within five years before the survey were interviewed for ANC visit and place of birth. The number of ANC visit was measured as in number (0–20) and place of delivery were collected as respondents home, relatives home, and health institutions. We analyze the 4,693 women data that give birth within the survey after antenatal booking.

Maternal continuum of care is a series of care of mothers from pregnancy until the postnatal period including 4 and above ANC, skilled delivery, and at least one postnatal checkup [19]. Dropout from maternity continuum of care: could be one of the following: not having 4 and above ANC, delivery out of health facility, and not having a postnatal checkup. Dropout from antenatal care visit was considered if women had no at least 4 ANC visits after booking for ANC service. Dropout from institutional delivery was considered if women give birth out of health institution after antenatal care booking. Dropout from PNC was considered if a woman doesn't have any PNC visit after she gives the most recent birth. A complete continuum of care was defined as if women had four or above ANC visits, institutional delivery, and postnatal care visit.

## Data analysis

After the data was accessed from the major DHS program; data cleaning, recoding, and weighting was done using Stata 14. The data were weighted using the women weighting variable (V005) as per the recommendation of the major DHS program. We use Survey (svy) command for descriptive and analytical analysis. The detail of how to weigh the data found from the EDHS 2016 report Annex A[3]. Descriptive findings were reported using narratives, figures, and tables. A binary logistic regression model was fitted for the three outcomes (dropout from ANC, Institutional delivery, and PNC services). Chi-square was checked and bi-variable analysis was done for all the variables listed in the descriptive table. Variables with p-value<0.2 were used for multivariable analysis. The multicollinearity assumption was tested using a variance inflation factor; place of residence and respondents education had multi-collinearity. Model goodness of fit was tested using Hosmer and Lemeshow goodness of fit test for all the tree models. Finally, AOR with 95% CI were reported for all variables in the multivariable analysis with p-value <0.05 as a cut of point to determine statistically significant determinant factors.

## Ethics approval and consent to participate

Permission for data access was obtained from the Major DHS program (http://www.dhsprogram.com) after registered as an authorized user. All the data used for this manuscript are publically available and confidentiality was maintained anonymously.

## Results

Among 4, 693 study participants, 3,303 (70.38%) were rural residents. Of rural residents, 1589 (48.11%) had four and above ANC visits, 1,305 (39.51%) institutional delivery, and 224(6.78%) postnatal care visits. From those who report distance from health facility as a big problem 1,041(50.05%) had four and above ANC visits, 824 (39.62%) institutional delivery, and 119 (5.72) postnatal care visits (Table 1).

Among women who booked for ANC only 308 (6.56%) used the complete continuum of care. Among women who had a recommended ANC (four and above) 11.84% used the complete continuum of care. Among the women who were booked for ANC, the dropout rate was 10.85% and among those who give birth from the health institution after 4 and above ANC visit, 779 (29.96%) dropped from institutional delivery. Moreover, among those women who had four and above ANC and give birth from the health institutions, 1,382 (81.78%) were dropped from postnatal care service (Table 2).

We found a high dropout proportion from institutional delivery and postnatal care visits after having four and above ANC visits. The magnitude of the dropout rate varies across regions; the highest dropout rate from institutional delivery was observed in the Afar region (57.55%) and lowest in Addis Ababa (3.88%). Similarly, the highest and lowest dropout rate from postnatal checkup was found in Harari (97.96%) and (74.93%), respectively (Table 3).

### Risk factors for dropout from maternity continuum of care

Several socio-economic, socio-demographic, and reproductive health factors were significantly found to affect dropout from the three maternal continuum of care after antenatal care booking.

Not being informed about pregnancy complications during women's ANC follow up was the only variable significantly associated with all the three dropout rates from the continuum of care. Women who were not informed about pregnancy complications had nearly two times

**Table 1. Background characteristics of reproductive-age women who give birth within five years before the survey after antenatal care booking in Ethiopia, EDHS 2016 perspective.**

| Variable name | | Number of antenatal care | | Place of delivery | | Postnatal care checkup | |
|---|---|---|---|---|---|---|---|
| | | 1–3 | 4 and above | Out of health institution | Health institution | No | Yes |
| Place of residence | Urban | 378(27.19%) | 1072 (72.81%) | 185(13.31%) | 1,205(86.69%) | 1,218 (87.63%) | 172 (12.37%) |
| | Rural | 1714 (51.89%) | 1589 (48.11%) | 1,998(60.49%) | 1,305(39.51%) | 3,079 (93.22%) | 224(6.78%) |
| Religion | Protestant | 401(47.01%) | 452(52.99%) | 472(55.33%) | 381(44.67%) | 802(94.02%) | 51(5.98%%) |
| | Orthodox | 669(36.20%) | 1,179 (63.80%) | 656 (35.50%) | 1,192(64.50%) | 1,630 (88.20%) | 218 (11.80%) |
| | Muslim | 993(51.50%) | 935(48.50%) | 1,019(52.85%) | 909(47.15%) | 1,804 (93.57%) | 124(6.43%) |
| | Other | 29(45.31%) | 35(54.69%) | 36(56.25%) | 28(43.75%) | 61(95.31%) | 3(4.69% |
| Current marital status | Not married | 145(35.19%) | 267(64.81%) | 139(33.74%) | 273(66.26) | 360(87.38%) | 52(12.62%) |
| | Married | 1,947 (45.48%) | 2,334 (54.52%) | 2,044 (47.75%) | 2,237 (52.25%) | 3,937 (91.96%) | 344(8.04%) |
| Current occupation | Not working | 1,473 (47.06%) | 1,657 (52.94%) | 1,570(50.16%) | 1,560 (49.84%) | 2,913 (93.07%) | 217(6.93%) |
| | Working | 619 (39.60%) | 944(60.40%) | 613(39.22%) | 950(60.78%) | 1,384 (88.55%) | 179 (11.45%) |
| Highest education level | No education | 1,255 (53.93%) | 1,072 (46.07%) | 1,470 (63.17) | 857(36.83) | 2,181(93.73) | 146(6.27) |
| | Primary | 641 (41.65%) | 898(58.35%) | 623(40.48%) | 916(49.52%) | 1,400 (90.97%) | 139(9.03%) |
| | Secondary | 133 (25.63%) | 386 (74.37%) | 73(14.07%) | 446(85.93%) | 453 (87.28%) | 66(12.72%) |
| | Higher | 63 (20.45%) | 245(79.55%) | 17(5.52%) | 291(94.48%) | 263(85.39%) | 45(14.61%) |
| Wealth index | Poorest | 655(61.56%) | 409(38.44%) | 763(71.71%) | 301(28.29%) | 1,011 (95.02%) | 53(4.98%) |
| | Poorer | 390 (51.32%) | 370(48.68%) | 442 (58.16%) | 318(41.84%) | 702(92.37%) | 58(7.63%) |
| | Middle | 355 (50.64%) | 346(49.36%) | 409(58.35%) | 292(41.65%) | 649(92.58%) | 52(7.42%) |
| | Richer | 290 (43.35%) | 379 (56.65%) | 347(51.87%) | 322(48.13%) | 613(91.63%) | 56(8.37%) |
| | Richest | 402 (26.82%) | 1,097 (73.18%) | 222(14.81%) | 1,277(85.19%) | 1,322 (88.19%) | 177 (11.81%) |
| Media exposure | No | 1,368 (53.71%) | 1,179 (46.29%) | 1,564(61.41%) | 983(38.59%) | 2,390 (93.84%) | 157(6.16%) |
| | Yes | 724(33.74%) | 1,422 (66.26%) | 619(28.84%) | 1,527(71.16%) | 1,907 (88.86%) | 239 (11.14%) |
| Distance to the health facility | Big problem | 1,039 (49.95%) | 1,041 (50.05%) | 1,256(60.38%) | 824(39.62%) | 1,961 (94.28%) | 119(5.72%) |
| | Not big problem | 1,053 (40.30%) | 1,560 (59.70%) | 927(35.48%) | 1,686(64.52%) | 2,336 (89.40%) | 277 (10.60%) |
| Birth order | 1 (primipara) | 432(37.70%) | 714(62.30%) | 325(28.36%) | 821(71.64%) | 1,015 (88.57%) | 131 (11.43%) |
| | 2 to 4 parity | 929(43.47%) | 1,208 (56.53%) | 976(45.67%) | 1,161(54.33%) | 1,965 (91.95%) | 172(8.05%) |
| | Five and more | 731(51.84%) | 679 (48.16%) | 882(62.55%) | 528(37.45%) | 1,317 (93.40%) | 93(6.60%) |
| Wanted pregnancy when becoming pregnant | Then | 1,678 (44.76%) | 2,071 (55.24%) | 1,755(46.81%) | 1,994(53.19%) | 3,440 (91.76%) | 309(8.24%) |
| | Later | 296(42.71%) | 397(47.29%) | 311(48.88%) | 382(51.12%) | 629(90.76%) | 64(9.24%) |
| | No more | 118(47.01%) | 133(52.99%) | 117(46.61%) | 134(53.39%) | 228(90.84%) | 23(9.16%) |

(*Continued*)

**Table 1.** (Continued)

| Variable name | | Number of antenatal care | | Place of delivery | | Postnatal care checkup | |
|---|---|---|---|---|---|---|---|
| | | 1–3 | 4 and above | Out of health institution | Health institution | No | Yes |
| Age of respondent | 20 to 34 | 1,502 (43.97%) | 1,914 (56.03%) | 1,555(45.52%) | 1,861(54.48%) | 3,134 (91.74%) | 282(8.26%) |
| | 15 to19 | 127(50.80%) | 123 (49.20%) | 103(41.20%) | 147(58.80%) | 232(92.80%) | 18(7.20%) |
| | 35 and above | 463(45.08%) | 564(54.92%) | 525(51.12%) | 502(48.88%) | 931(90.65%) | 96(9.35%) |
| Told about pregnancy complications during the ANC visit | Yes | 736 (35.18%) | 1,437 (53.25%) | 789(36.14%) | 1,384(55.14%) | 1,777 | 396 (65.24%) |
| | No | 1,356 (64.82%) | 1,164 (44.75%) | 1,394(63.86%) | 1,126(44.86%) | 2,309 (56.51%) | 211 (34.76%) |
| Covered by health insurance | Yes | 71(3.39%) | 144 (5.54%) | 67(3.07%) | 148(5.90%) | 175(4.28%) | 40 (6.59%) |
| | No | 2,021 (96.61%) | 2,457 (94.46%) | 2,116 (96.93%) | 2,362 (94.10%) | 3,911 (95.72%) | 567 (93.41%) |

*Media exposure*: Media exposure was calculated from the internet use, TV watching, radio listening, reading newspapers and those who score above the median were considered as having media exposure and the rest considered as having no media exposure.

(AOR = 1.80, 95%CI: 1.49, 2.18), (AOR = 2.13, 95%CI: 1.75, 3.05), (AOR = 1.58, 95%CI: 1.26, 1.93) more odds of dropout from ANC, PNC, and institutional delivery respectively compared to their counterpart. Women who were from the poorest family had one and a half (AOR = 1.71, 95%CI: 1.12, 2.61) more odds of dropout from antenatal care visits compared to women from richest families. Women who were not married had 42% (AOR = 0.58, 95% CI: 0.38, 0.88) fewer odds of dropout from antenatal care visits compared to those who were married.

Moreover, religion, distance from the health facility, wealth index, being informed about pregnancy complications, media exposure, and birth order were significantly associated with drop out of institutional delivery (Table 4).

## Discussion

In this study, we have tried to assess the magnitude and associated factors of dropout from the recommended number of ANC, institutional delivery, and postnatal care visit using the EDHS 2016 data set.

The dropout rate from maternity continuum of care was high compared to other studies conducted in Nigeria of which women 38.1% and 50.8% of the women who receive ANC were dropped out from skilled delivery and PNC respectively [18], Cambodia About 90% had at least one ANC, 60% of them 4 and above, 74% Skill birth attendance and 71% had at least one postnatal checkup[13], and Debremarkos where women had 32.2% dropout from all continuum of care, while 66.4% and 84.1% had four and above ANC, and institutional delivery respectively[15]. The difference might be due to variation in the population, quality, and

**Table 2. Summary of dropout proportion from each maternity continuum of care services in Ethiopia among antenatal care booked women (N = 4,693).**

| No of ANC Visit | | Dropout from institutional delivery | | Dropout from the postnatal visit | | Dropout from PNC after institutional delivery | | Dropout from PNC after out-of institution delivery | |
|---|---|---|---|---|---|---|---|---|---|
| | | Yes | No | No | Yes | No | yes | No | Yes |
| 1–3 Visit | 2092 | 1,272(60.80%) | 820(39.20%) | 194 (9.27%) | 1898 (90.73%) | 99(4.73%) | 721(34.46%) | 95(4.54%) | 1,177(56.26%) |
| ≥ 4 | 2601 | 911(35.02%) | 1690(64.98%) | 413(15.88%) | 2,188(84.12%) | 308(11.84%) | 1,382(53.13%) | 105(4.04%) | 806(30.98%) |

**Table 3. Regional variation of dropout proportion from a place of delivery and postnatal care visit among women who had four and above ANC (N = 2601).**

| Region | Dropout from institutional delivery | | Dropout from Postnatal care service after Four and above ANC | | Dropout from PNC after having institutional delivery | |
|---|---|---|---|---|---|---|
| | Yes (%) | No (%) | Yes (%) | No (%) | Yes (%) | No (%) |
| Tigray | 95(22.04) | 336(77.96) | 333(77.26) | 98(22.74) | 260(77.38) | 76(22.62) |
| Afar | 61(57.55) | 45(42.45) | 93(87.74) | 13(12.26) | 35(77.78) | 10(22.22) |
| Amhara | 119(51.52) | 112(48.48) | 192(83.12) | 39(16.88) | 89(79.46) | 23(20.54) |
| Oromia | 128(55.17) | 104(44.83) | 212(91.38) | 20(8.62) | 90(86.54) | 14(13.46) |
| Somali | 48(47.52) | 53(52.48) | 97(96.04) | 4(3.96) | 51(96.23) | 2(3.77) |
| Benishangul Gumuz | 109(47.19) | 122(52.81) | 181(78.35) | 50(21.65) | 88(72.13) | 34(22.87) |
| SNNPR | 176 (50.14) | 175(49.86) | 311(88.60) | 40(11.40) | 150(85.71) | 25(14.29) |
| Gambela | 67(36.22) | 118(63.78) | 154(83.24) | 31(16.76) | 96(81.36) | 22(18.64) |
| Harari | 33(22.45) | 114(77.55) | 144(97.96) | 3(2.04) | 112(98.25) | 2(1.75) |
| Addis Ababa | 13(3.88) | 322(96.12) | 251(74.93) | 84(25.07) | 242(75.16) | 80(24.84) |
| Dire Dawa | 62(24.70) | 189(75.30) | 220(87.65) | 31(12.35) | 169(89.42) | 20(10.58) |

ANC: Antenatal Care, PNC: Post Natal Care, SNNPR: Southern Nations Nationalities and Peoples Region,

accessibility of health facilities which could give pregnancy and child health services. Specifically, the study in Debremarkos was among urban residents and the accessibility and quality of health care services and facilities were better in urban areas than the general population [20]. The dropout rate was comparable with other studies conducted in Khammouane which was a 10% complete continuum of care, 30.8% used PNC, and 29.7% institutional delivery [6].

Unmarried women had nearly 50% (AOR = 0.58, 95%CI: 0.38, 0.88) fewer odds of dropout from antenatal care service after being registered for it compared to married. On the contrary, some studies conducted elsewhere have noted that marital status was not significantly associated with dropout from the maternity continuum of care [6, 7, 12, 13, 15, 18, 19]. This might be due to the effect of reduced women's autonomy on decision making. Especially in developing countries including Ethiopia, [12] husbands/partners are decision-makers in many aspects of the family issues including reproductive health services utilization. As a result, pregnant women might have forced to miss their ANC visits and other reproductive health services. Cognizant to this, In Sub-Saharan Africa, women who had 4 and above ANC visits had more autonomy on decision making than others [16]. Another study in Pakistan reported that women who had the autonomy to decide on health care seeking behavior had better utilization of maternity continuum of care [12].

Wealth index was one of the important predictors of dropout from the maternity continuum of care. Accordingly, women who were from the poorest wealth index had 71% (AOR: 1.71, 95%CI, 1.12–2.61) more odds of dropout from ANC visit compared to those women from the richest family. This result was supported by other studies conducted in Nigeria [18], Tanzania, [12], and Cambodia [13] which showed that poor wealth index was the main determinant factor for not having recommended number of ANC visit. This could be justified in that women from poor wealth index might be low socioeconomic status decreased health service utilization, limited access to, and quality of services [2, 11, 13].

Women who had no media exposure had fewer odds of institutional delivery compared to those who had media exposure. This finding was supported by other studies conducted in Ethiopia [21], Pakistan [22], and Asia [23]. This might be since women who have media exposure might have better knowledge about the importance of institutional delivery and may create a positive attitude to give birth at health institutions [24, 25].

**Table 4. Factors associated with a dropout of maternity continuum of care among reproductive-age women in Ethiopia, 2016 EDHS perspective.**

| Variable name | | Dropout from | | |
|---|---|---|---|---|
| | | ANC | institutional delivery | postnatal care visit |
| | | AOR (95%CI) | AOR (95%CI) | AOR (95%CI) |
| Religion | Protestant | 1.08(0.80,1.46) | 1.56 (1.12,2.18)* | 1.85 (1.20,2.84)* |
| | Orthodox | Ref | Ref | Ref |
| | Muslim | 1.10(0.85,1.43) | 1.24(0.92,1.67) | 1.22 (0.86,1.74) |
| | Other | 1.82(096,3.45) | 2.98 (1.56, 5.71)* | 5.30 (0.93,30.48) |
| Current marital status | Not married | 0.58(0.38,0.88)** | Not significant | Not significant |
| | Married | Ref | Not significant | Not significant |
| Current occupation | Not working | Not significant | Not significant | 0.85(0.63, 1.15) |
| | Working | Not significant | Not significant | Ref |
| Wealth index | Poorest | 1.71(1.12,2.61) * | 7.12(4.70,10.78) * | 1.44(0.78,2.65) |
| | Poorer | 1.38(0.88,2.18) | 4.86(3.42,6.89) * | 1.74(0.97,3.13) |
| | Middle | 1.43(0.94,2.18) | 4.88(3.51,6.77) * | 1.06(0.60, 1.87) |
| | Richer | 1.12(0.73,1.69) | 4.31(3.23,5.74) * | 0.90(0.50,1.62) |
| | Richest | Ref | Ref | Ref |
| Media exposure | No | 1.09(0.87,1.36) | 1.38(1.10, 1.74) * | 1.28(0.91,1.81) |
| | Yes | Ref | Ref | Ref |
| Distance to a health facility | Big problem | 0.99(0.81,1.23) | 1.41(1.10,1.81) * | 1.31(0.96,1.79) |
| | Not a big problem | Ref | Ref | Ref |
| Birth order | 1 (primipara) | Ref | Ref | Ref |
| | 2 to 4 parity | 0.93(0.71,1.22) | 2.34(1.79,3.06) * | 1.18(0.87,1.59) |
| | Five and more | 0.96(0.72,1.29) | 4.10(2.88,5.85) * | 0.97(0.69,1.36) |
| Age in years of respondents | 20 to 34 | Ref | Ref | Not significant |
| | 15 to19 | 1.27(0.80,2.02) | 1.22(0.79,1.87) | Not significant |
| | 35 and above | 0.92(0.70,1.21) | 0.75(0.54,1.05) | Not significant |
| Being informed about pregnancy complications | Yes | Ref | Ref | Ref |
| | No | 1.80(1.49,2.18) * | 1.58(1.26,1.93) * | 2.31(1.75,3.05) * |
| Covered by health insurance | No | Ref | Ref | Ref |
| | Yes | 0.80(0.57,1.12) | 0.77(0.49,1.21) | 1.01(0.61,1.68) |

CI: Confidence Interval, Ref: Reference category, Not significant: Not significant in the bi-variable analysis.

**represents negatively associated factors

*represents positively associated

Birth order was another variable significantly associated with the dropout rate. Hence, we found that increasing birth order increases the odds of dropout from institutional delivery. This finding was in agreement with a study conducted in rural Tanzania [26], where nulliparous women were more likely to give birth at health institutions than multiparous women. This might be justified by nulliparous women who were very sensitive to pregnancy-related complications and prefer to give birth at health institutions.

Coming to institutional delivery, women living in a community where distance was a big problem had nearly one and a half times more odds of giving birth out of health institutions. Our finding is supported by other studies Lao People's Democratic Republic [6], India [27] and Ethiopia [20] distance from health facilities which could provide delivery service reduces the utilization of maternal continuum of care. Accessibility is an important determinant factor in the utilization of maternity care services [28]. This might be women from places where the

distance to a health facility as a big problem may have difficulty to arrive in the health facility and leads to delay 2(delay in arriving in health facilities).

Most consistently, we found that information given for pregnant women about pregnancy complications during their ANC visit decreases the dropout rate from ANC, skilled delivery, and postnatal care visits. Women who were not informed about pregnancy complications during their ANC visit had 1.8, 1.5, and more than 2 times more odds of dropout from ANC, skilled delivery, and PNC as compared to their counterparts. Supported by a study conducted in Cambodia where those pregnant women informed about signs of complication had better odds of continuing maternity care[13].This might be supported by different studies which point out that the quality and frequency of ANC services [8, 9, 19, 29], and having any complications during pregnancy [19] decreases the dropout rate of maternity continuum of care.

## Limitations

We are confident that our research was strong but not immune to limitations. As it was secondary data we can't include health service quality and accessibility explicitly. Since the study was based on women who give birth within five years before the survey, it may introduce recall bias.

## The implication of the study

It may help policymakers by showing the dropout rate from each maternity continuum of care and the determinant factors associated with the dropout rate.

## Conclusion and recommendations

The dropout rate from the maternity continuum of care was high in Ethiopia. Different socioeconomic, pregnancy and health service-related variables were significantly associated factors of high dropout rate from each continuum of care among ANC booked women. We recommend the Ethiopian Ministry of Health and other stakeholders to give more emphasis to improve the maternity continuum of care by reducing the identified factors.

## Acknowledgments

The authors are happy to acknowledge the Institute of Public Health, College of Medicine and Health Science, University of Gondar. Our thanks also extend to the international Major DHS program for permitting data access.

## Author Contributions

**Conceptualization:** Atalay Goshu Muluneh, Mehari Woldemariam Merid.

**Data curation:** Atalay Goshu Muluneh, Geta Asrade Alemayehu.

**Methodology:** Atalay Goshu Muluneh, Getahun Molla Kassa, Geta Asrade Alemayehu, Mehari Woldemariam Merid.

**Validation:** Getahun Molla Kassa, Geta Asrade Alemayehu, Mehari Woldemariam Merid.

**Writing – original draft:** Atalay Goshu Muluneh, Mehari Woldemariam Merid.

**Writing – review & editing:** Getahun Molla Kassa, Geta Asrade Alemayehu, Mehari Woldemariam Merid.

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
