## [Decision Letter · Decision Letter 0]

28 Feb 2020

PONE-D-20-01331

High dropout rate from maternity continuum of care after antenatal care booking and its associated factors among reproductive age women in Ethiopia, Evidence from Demographic and Health Survey 2016

PLOS ONE

Dear Mr Muluneh,

Thank you for submitting your manuscript to PLOS ONE. After careful consideration, we feel that it has merit but does not fully meet PLOS ONE’s publication criteria as it currently stands. Therefore, we invite you to submit a revised version of the manuscript that addresses the points raised during the review process.

We would appreciate receiving your revised manuscript by Apr 13 2020 11:59PM. To enhance the reproducibility of your results, we recommend that if applicable you deposit your laboratory protocols in protocols.io, where a protocol can be assigned its own identifier (DOI) such that it can be cited independently in the future. For instructions see: http://journals.plos.org/plosone/s/submission-guidelines#loc-laboratory-protocols

We look forward to receiving your revised manuscript.

Kind regards,

Bruce A Larson

Academic Editor

PLOS ONE

Journal Requirements:

1.

We suggest you thoroughly copyedit your manuscript for language usage, spelling, and grammar. If you do not know anyone who can help you do this, you may wish to consider employing a professional scientific editing service.  

2. Thank you for including your funding statement; "No funding organization "

Please provide an amended Funding Statement that declares *all* the funding or sources of support received during this specific study (whether external or internal to your organization) as detailed online in our guide for authors at http://journals.plos.org/plosone/s/submit-now.  

Please state what role the funders took in the study.  If any authors received a salary from any of your funders, please state which authors and which funder. If the funders had no role, please state: "The funders had no role in study design, data collection and analysis, decision to publish, or preparation of the manuscript."

2. Please include your tables as part of your main manuscript and remove the individual files. Please note that supplementary tables (should remain/ be uploaded) as separate "supporting information" files

Additional Editor Comments (if provided):

Dear Author's,

Both reviewers recommend that a "major revision" is needed.

Please respond carefully to each reviewer, especially reviewer 2.

Both reviewers also recommend that the manuscript undergoes major editing to improve grammar, presentation, flow, etc.

Reviewers' comments:

Reviewer's Responses to Questions

**Comments to the Author**

1. Is the manuscript technically sound, and do the data support the conclusions?

Reviewer #1: Yes

Reviewer #2: Partly

2. Has the statistical analysis been performed appropriately and rigorously? 

Reviewer #1: Yes

Reviewer #2: I Don't Know

3. Have the authors made all data underlying the findings in their manuscript fully available?

Reviewer #1: Yes

Reviewer #2: Yes

4. Is the manuscript presented in an intelligible fashion and written in standard English?

Reviewer #1: No

Reviewer #2: No

5. Review Comments to the Author

Reviewer #1: Abstract

The second sentence under “methods” should be deleted. Authors had initiated stated that they conducted secondary data analysis, therefore, the second sentence is not necessary

In the results, they should present dropout rate for each indicator of maternal care. This becomes necessary because there were results for factors associated with each of them.

Main manuscript

This section lacks coherence. Although, the authors tried to justify the paper by making reference to high level of childhood and maternal mortality, they failed to situate the work within the larger body of knowledge in the subject area. It’s difficult to know what new knowledge is been added o maternal care broadly and specifically in Ethiopia.

In the Methods section, authors might want to provide richer contextual information about maternity care in Ethiopia and how it has fared over time.

The suggested recommendation should be more specific and linked to the results of the study.

Reviewer #2: The authors presented a secondary analysis of Demographic and Health Survey data for women of reproductive age in Ethiopia, assessing dropout rates and associated factors along the continuum of care for maternal health. I have a few comments that could help clarify the approach and findings of the study. In general, I also recommend that the manuscript undergoes major editing to improve grammar and presentation, as well as the flow and clarity of ideas.

Title: I feel the title can be more concise. Having read the manuscript, perhaps phrases like “after antenatal care booking” can be left out of the title?

Abstract

Objective: Please restate objective to be more concise. For example, is it associated factors of continuum of care or associated factors on/along continuum of care?

Methods

• I am not clear on the weighting approach, and whether it refers to the original survey or the study being reported.

Results

• What are the factors associated with completing the whole continuum? Would be interesting to see if there are any significant differences between the non-completed and completed groups.

• I feel factors associated with dropout from delivery care could be stated more concisely with some editing

• Being informed or LACK of information on pregnancy complications from ANC was associated with ANC, delivery and PNC?

Conclusion

• Perhaps less repetition of results (The 6.56% figure) and add a sentence on what the study recommends?

Background

Line 57-58 maybe a sentence or two to give reader idea of the extent of maternal health challenge in Ethiopia, such as mortality rate and some of the specific challenges faced?

A short description of maternity continuum of care as it is conceptualized in this study can be included in the background.

Line 69-71 I feel that more precise/objective terms can be used here – instead of “lion’s share” perhaps an exact statistic to give reader an idea of the magnitude of the problem.

Methods

Line 88 – was the study representative at the national level?

The study design section should include information about study design of this particular study, not just the original survey

What is the difference between “7,589 women who give birth within five years before the survey were interviewed for ANC visit and place of birth” and the “4,693 women data that give birth within the survey after antenatal booking”. Is one a sub-sample of another? Is the sample of 4693 also representative nationwide?

Results

Line 134-136 also please offer a contrast with urban residents – how many attended ANC, institutional delivery etc.

Line 143 “Among 4,693 women, only 308 (6.56%) or 11.84% from those having four and above ANC visit have had complete maternity continuum of care”. This statement is confusing, I am not sure what the finding is.

“among those who give birth from the health institution 145 after 4 and above ANC visit, 779 (29.96%) dropped from institutional delivery” How do people who gave birth in institution drop out from institutional delivery?

Line 144 “Among the women who were booked for ANC, the dropout rate was 10.85%” Drop out from what, antenatal care?

Perhaps for precision and to write more concisely, phrases like “4 and above ANC visit” can be shortened to “completed ANC”. It helps when there is a long sentence describing findings, the reader doesn’t get confused

In the method the authors described what drop out from the continuum meant for each stage, but not what the dropout rate is? How it is defined. And it should be stated clearly more than one drop out rate is considered by referring to drop out rateS, perhaps even in the title.

Please adjust Table 2 to look neater – that is, not cut out short words such as “visit”

Table 2 : It is not clear to me what these proportions are. If drop out from institutional delivery for the category >4 visits is 65%, does that mean those women delivered at home? And how does one conceptualize a 35% drop out from home delivery then? Drop out from home delivery was not even specified as a variable of interest in the methods section– it does not constitute part of the definition of continuum of care in this study. Other variables such as dropout from “No PNC and institutional delivery” are also confusing. How does one drop out from “No PNC”? I think the authors should clarify with a modified title for Table 2 and a legend for Table 2 to really clarify what these proportions mean.

Line 152: “We found a high dropout rate from institutional delivery and postnatal care visit after having four 152 and above ANC visit”. Again, it is difficult to read this from Table 2 because I am not sure if the authors are looking at PNC among all – yes/no categories. I am not sure if we are looking at the 15.8% as proportion of those who DID drop out, or as those who HAD postnatal care. The table title says dropout proportion but as stated above, it is presented in a quite ambiguous manner. And the authors should be consistent with the use of proportion vs dropout rate vs dropout proportion. Again, a legend to Table 2 will help.

It could help if Table 2 is presented like Table 3, making it clear the dichotomy of drop-out vs no drop-out, compared against the >4 and <4 ANC visit groups.

In Table 3, please report the total number from each region.

Please refer to the Table numbers in the text when reporting the results, to make it easier for reader to follow.

Line 164 Being informed or NOT being informed. It seems in 166 the authors state that NOT being informed was associated with drop out. This should be clear in the abstract and the Line 164 to avoid confusion of results.

Please do a legend for Table 4 to shed more light on variables such as media exposure and the wealth index.

Discussion

Line 184-185 might be useful to give one or 2 examples of the rates from these other contexts to make the comparison much clearer.

Line 185 please specify Debremarkos, Ethiopia since the other two are countries.

Line 187 what were the rates in Debremarkos?

Line 189-90 what was the rate?

Line 194-195 if the authors are to qualify the assertion about decision making, they have to compare some of the contexts in the studies cited that found no significant association with Ethiopia.

I hope the comments will be clear and useful to the authors for revising the manuscript.

6. PLOS authors have the option to publish the peer review history of their article (what does this mean?). If published, this will include your full peer review and any attached files.

Reviewer #1: No

Reviewer #2: No

---

## [Author Response · Author response to Decision Letter 0]

19 Mar 2020

Author's response to reviews 

High dropout rate from maternity continuum of care after antenatal care booking and its associated factors after antenatal care booking, Evidence from Demographic and Health Survey 2016 (PONE-D-20-01331)

Atalay Goshu Muluneh*1, Getahun Molla Kassa1, Geta Asrade Alemayehu2, Mehari W/Mariam Merid1

1Department of Epidemiology and Biostatistics, Institute of Public Health, College of Medicine and Health Sciences, University of Gondar, Gondar, Ethiopia

2Department of Health System and Policy, Institute of Public Health, College of Medicine and Health Sciences, University of Gondar, Gondar, Ethiopia

*Corresponding author: 

Atalay Goshu Muluneh 

Email addresses:

AG: goshuatalay12@gmail.com

GM: getahunm08@gmail.com

GA: getasrade64@gmail.com

 MWM: mehariho19@gmail.com

Postal address: P.O. Box 196, Gondar, Ethiopia

Version 1, Date: April, 2020

With regards! 

From: Atalay Goshu Muluneh 

Correspondence Author

Author's response to reviews: Find it below

To: PLOS ONE, editorial office 

Subject: Submitting a revised version of a manuscript 

Object: High dropout from maternity continuum of care after antenatal care booking and its associated factors among reproductive age women in Ethiopia, Evidence from Demographic and Health Survey 2016 (PONE-D-20-01331)

We would like to thank the reviewers and editor for sharing the view and experience. The comments are very important that will improve the manuscript. The point-by-point responses for each of the comments and the revised manuscript are provided in the attached documents. 

POINT BY POINT RESPONSES

We would like to take this opportunity to thank the reviewers and the editor for sharing their view and constructive comments. The point-by-point responses for each of the comments are provided in the following pages.

Authors’ response of reviewer 1 

REVIEWER 1 COMMENT AUTHORs RESPONSE

1. Abstract

The second sentence under “methods” should be deleted. Authors had initiated stated that they conducted secondary data analysis, therefore, the second sentence is not necessary

In the results, they should present dropout rate for each indicator of maternal care. This becomes necessary because there were results for factors associated with each of them. Thank you very much!

Thank you and we update based your recommendation. We add the dropout rate “2,092(44.58%), 2,183 (46.52%) and 4,086(87.07%) dropped from recommended number of ANC, Institutional delivery and postnatal care visit respectively.” See line 44-46.

2. This section lacks coherence. Although, the authors tried to justify the paper by making reference to high level of childhood and maternal mortality, they failed to situate the work within the larger body of knowledge in the subject area. It’s difficult to know what new knowledge is been added o maternal care broadly and specifically in Ethiopia.

In the Methods section, authors might want to provide richer contextual information about maternity care in Ethiopia and how it has fared over time.

The suggested recommendation should be more specific and linked to the results of the study.

 Thank you for your great insight and we acknowledge the gaps.

o We try to rearrange 

o The 1st paragraph of the background is about status of maternal health and continuum of care in Ethiopia. see line 68-84

o We update the recommendation based on specific findings. See line 58-61

3. The manuscript was not presented in an intelligible fashion and written in standard English

 Thank You!

Accepted and it has been given for a local language editor and we update based on his edition. 

Here is the professional Experience of the local language editor:

Name of the Editor: Demeke W.Ghiorges

Age: 72 years

Academic degrees: BA in English language and literature, Addis Ababa University, 1983, M.A. teaching English as a foreign language, Addis Ababa university 1991

Academic rank: Assistant professor in English

Teaching experience: over 30 years

Editing Experience: Language editor of the Journal of Medicine and Biomedical Sciences, College of Medicine and Health Sciences, the University Of Gondar, Ethiopia

Telephone: 0918034043

Reviewer 2 

REVIEWER COMMENT AUTHORs RESPONSE

1. I feel the title can be more concise. Having read the manuscript, perhaps phrases like “after antenatal care booking” can be left out of the title? Thank you for your valuable comment!

Accepted:

See the title page line 1 and line 35 of the abstract section

2. Abstract

Objective: Please restate objective to be more concise. For example, is it associated factors of continuum of care or associated factors on/along continuum of care? Thank you!

o We update the objectives based on your comment and see line 35 of the abstract

3. Absttract: Methods

• I am not clear on the weighting approach, and whether it refers to the original survey or the study being reported. Thank you in advance!

• The survey recommends weighting the data for any inferential statistics. “Due to the non-proportional allocation of the sample to different regions and their urban and rural areas and the possible differences in response rates, a sampling weight must be used in all analyses using the 2016 EDHS data to ensure the actual representative of the survey results at both the national and domain levels(1)” . Weighting the data is a must to balance the above mentioned differences in response rate, and non-proportional allocations. Weighting variables are also available for each data set and we use the recommended weighting variable for women data. See the detail from the main Ethiopian Demographic and health Survey 2016 report Appendix A4. Sampling Weight on page 337. 

4. Abstract results, what are the factors associated with completing the whole continuum? Would be interesting to see if there are any significant differences between the non-completed and completed groups.

• I feel factors associated with dropout from delivery care could be stated more concisely with some editing

• Being informed or LACK of information on pregnancy complications from ANC was associated with ANC, delivery and PNC?

 Thank you,

We are interested to find factors of why women dropout from maternity continuum of care. We fit three models for recommended number of ANC visit (Four and above ANC), Institutional delivery and postnatal care. Information given about complications of pregnancy during their ANC visit was significantly associated with all continuums of cares (from recommended ANC, Skilled delivery and PNC) but other variables were specific for all continuums of care. 

5. Abstract, Conclusion

• Perhaps less repetition of results (The 6.56% figure) and add a sentence on what the study recommends?

 Acknowledged! 

We update the conclusion based on your recommendation and amended objective. See line 54-59.

6. Line 57-58 maybe a sentence or two to give reader idea of the extent of maternal health challenge in Ethiopia, such as mortality rate and some of the specific challenges faced?

A short description of maternity continuum of care as it is conceptualized in this study can be included in the background. Thanks,

According to the EDHS 2016 report, Ethiopian women had:

o Ethiopian women had a 21 per 1,000 women life time risk of death related to pregnancy 

o a maternal death of 412 per 100,000 live births

See line 75-77 of the background 

7. Line 69-71 I feel that more precise/objective terms can be used here – instead of “lion’s share” perhaps an exact statistic to give reader an idea of the magnitude of the problem.

 Thank you,

There are a number of literatures stating the importance of maternal continuum of care to reduce maternal and child mortality. For example 65% of the maternal deaths and 75% of maternal deaths occur in the early postnatal period which could be prevented by proper postnatal care services. In Ethiopia if a timely and comprehensive postnatal care service was given, neonatal mortality could be reduced by 10-17% (2).

8. Methods

Line 88 – was the study representative at the national level?

The study design section should include information about study design of this particular study, not just the original survey.

What is the difference between “7,589 women who give birth within five years before the survey were interviewed for ANC visit and place of birth” and the “4,693 women data that give birth within the survey after antenatal booking”. Is one a sub-sample of another? Is the sample of 4693 also representative nationwide?

 Thank you very much,

The data was representative at national level. Samples were taken from all regions including urban, rural and other pastoralist communities with large sample size. The major demographic and Health Survey reported as a standard and representative nationwide survey. We hope the 4,693 women who give birth after ANC booking are still representative. 

Sure, the original survey was a cross-sectional study and here we use the control study design approach. See line 109-110 of the methods section 

Other single studies try to address factors related to antenatal care booking. As a result, we are interested to determine factors why women lost from completing recommended ANC visit in Ethiopia after 1st ANC booking. So a total of 7,589 women give birth within five years before the survey. Among those, 4,693 of women were booked for ANC. And we use them as a study population. See line 129-134

9. Results

Line 134-136 also please offer a contrast with urban residents – how many attended ANC, institutional delivery etc. Thank you!

The detail is found in Table 1 as a 1st variable. Please see table 1st row. The table shows the ANC, Institutional delivery and PNC for each variables

10. Results: Line 143 “Among 4,693 women, only 308 (6.56%) or 11.84% from those having four and above ANC visit have had complete maternity continuum of care”. This statement is confusing, I am not sure what the finding is.

“among those who give birth from the health institution 

• Line 145 after 4 and above ANC visit, 779 (29.96%) dropped from institutional delivery” How do people who gave birth in institution drop out from institutional delivery?

 Thank you!

• Among women who booked for ANC only 308 (6.56%) used the complete continuum of care. Among women who had a recommended ANC (four and above) 11.84% used the complete continuum of care.

o From the line 145, as you know mothers who had a recommended number of ANC visit (Four and above) are expected to give birth at health institutions but 779(29.96%) of them give birth out of health institution. Those who give birth at the health institution were could not drop from institutional delivery. They already had institutional delivery. 

11. Results: Line 144 “Among the women who were booked for ANC, the dropout rate was 10.85%” Drop out from what, antenatal care?

Perhaps for precision and to write more concisely, phrases like “4 and above ANC visit” can be shortened to “completed ANC”. It helps when there is a long sentence describing findings, the reader doesn’t get confused

In the method the authors described what drop out from the continuum meant for each stage, but not what the dropout rate is? How it is defined. And it should be stated clearly more than one dropout rate is considered by referring to drop out rateS, perhaps even in the title.

Please adjust Table 2 to look neater – that is, not cut out short words such as “visit” Thank you!

10.85% dropout from complete (recommended ANC i.e. four and above)

The definition for dropout from rate from continuum of care was defined as proportion of women who had completed the recommended ANC among the study participants, women who give birth at the health institution, who had postnatal care visits. 

12. Results: Table 2 : It is not clear to me what these proportions are. If drop out from institutional delivery for the category >4 visits is 65%, does that mean those women delivered at home? And how does one conceptualize a 35% drop out from home delivery then? Drop out from home delivery was not even specified as a variable of interest in the methods section– it does not constitute part of the definition of continuum of care in this study. Other variables such as dropout from “No PNC and institutional delivery” are also confusing. How does one drop out from “No PNC”? I think the authors should clarify with a modified title for Table 2 and a legend for Table 2 to really clarify what these proportions mean.

 Thank you!

We try to update the table ( see table 2)

We consider dropout from institutional delivery as one variable of interest not dropout from the home delivery. 

In table 2 we try to describe the dropout rate from PNC from total ANC booked women, institutional delivery, and home delivery. Just to show how the institutional delivery is contributes for the increment of PNC utilization among reproductive age women. 

13. Results: Line 152: “We found a high dropout rate from institutional delivery and postnatal care visit after having four and above ANC visit”. Again, it is difficult to read this from Table 2 because I am not sure if the authors are looking at PNC among all – yes/no categories. I am not sure if we are looking at the 15.8% as proportion of those who DID drop out, or as those who HAD postnatal care. The table title says dropout proportion but as stated above, it is presented in a quite ambiguous manner. And the authors should be consistent with the use of proportion vs dropout rate vs dropout proportion. Again, a legend to Table 2 will help.

It could help if Table 2 is presented like Table 3, making it clear the dichotomy of drop-out vs no drop-out, compared against the >4 and <4 ANC visit groups.

In Table 3, please report the total number from each region.

Please refer to the Table numbers in the text when reporting the results, to make it easier for reader to follow.

 Thank you!

See table 2. 15.8% was those women who had PNC visit. The dropout rate was 84.2%.

The table three was about dropout from institutional delivery from all booked for ANC, and among those who had recommended number of ANC visit (four and above). And dropout from PNC. We try to update and use proportion instead of rate b/c the data was cross sectional and it is difficult to say rate rather proportion will be appropriate. See the abstract objective, table 2,3 and line 182.

14. Results: Line 164 Being informed or NOT being informed. It seems in 166 the authors state that NOT being informed was associated with drop out. This should be clear in the abstract and the Line 164 to avoid confusion of results.

Please do a legend for Table 4 to shed more light on variables such as media exposure and the wealth index.

 Thank you!

• Acknowledged and it is not being informed about pregnancy complications during their ANC visit. See line 52 of the abstract and line196 of the results.

• We make some variables like wealth index and put a legend for them below the table. See table 4 

15. Discussion

Line 184-185 might be useful to give one or 2 examples of the rates from these other contexts to make the comparison much clearer.

 Thank you!

The dropout rate from maternity continuum of care was high compared to other studies conducted in Nigeria of which women 38.1% and 50.8% of the women who receive ANC were dropped out from skilled delivery and PNC respectively (3), Cambodia (4), and Debremarkos where women had 32.2% dropout from all continuum of care, while 66.4% and 84.1% had four and above ANC, and institutional delivery respectively(5) . see line 217-223

16. Discussions, Line 185 please specify Debremarkos, Ethiopia since the other two are countries. Thank you!

The dropout rate from complete continuum of care was 32.2%, 66.4% had 4 and above ANC, 84.1% had institutional delivery. see line 220-221

17. Discussions: Line 187 what were the rates in Debremarkos? Thank you!

The dropout rate from complete continuum of care was 32.2%, 66.4% had 4 and above ANC, 84.1% had institutional delivery. see line 220-221

18. Discussions: Line 189-90 what was the rate? Thank you!

The magnitude was 10% complete continuum of care, 30.8% used PNC and 29.7% institutional delivery. See the line 227-228

19. Discussions: Line 194-195 if the authors are to qualify the assertion about decision making, they have to compare some of the contexts in the studies cited that found no significant association with Ethiopia. Thank you!

The context might be due to variation in the educational status of women, different socio-economic and cultural perspectives where women decision making ability is very important to reduce dropout from maternity continuum of care (6-9). 

1. Central Statistical Agency (CSA) [Ethiopia] and ICF. Ethiopia Demographic and Health Survey 2016. Addis Ababa, Ethiopia, and Rockville, Maryland, USA: CSA and ICF.

2. Wudineh KG, Nigusie AA, Gesese SS, Tesu AA, Beyene FY. Postnatal care service utilization and associated factors among women who gave birth in Debretabour town, North West Ethiopia: a community- based cross-sectional study. BMC pregnancy and childbirth. 2018;18(1):508.

3. Akinyemi JO, Afolabi RF, Awolude OA. Patterns and determinants of dropout from maternity care continuum in Nigeria. BMC pregnancy and childbirth. 2016;16(1):282.

4. Wang W, Hong R. Levels and determinants of continuum of care for maternal and newborn health in Cambodia-evidence from a population-based survey. BMC pregnancy and childbirth. 2015;15:62.

5. Amare NS, Araya BM, Asaye MM. Dropout from maternity continuum of care and associated factors among women in Debre Markos town, Northwest Ethiopia. bioRxiv. 2019:620120.

6. Iqbal S, Maqsood S, Zakar R, Zakar MZ, Fischer F. Continuum of care in maternal, newborn and child health in Pakistan: analysis of trends and determinants from 2006 to 2012. BMC health services research. 2017;17(1):189.

7. Ryan BL, Krishnan RJ, Terry A, Thind A. Do four or more antenatal care visits increase skilled birth attendant use and institutional delivery in Bangladesh? A propensity-score matched analysis. BMC public health. 2019;19(1):583.

8. Sakuma S, Yasuoka J, Phongluxa K, Jimba M. Determinants of continuum of care for maternal, newborn, and child health services in rural Khammouane, Lao PDR. PloS one. 2019;14(4):e0215635.

9. Singh K, Bloom S, Haney E, Olorunsaiye C, Brodish P. Gender equality and childbirth in a health facility: Nigeria and MDG5. African journal of reproductive health. 2012;16(3).

---

## [Decision Letter · Decision Letter 1]

5 May 2020

PONE-D-20-01331R1

High dropout rate from maternity continuum of care after antenatal care booking and its associated factors among reproductive age women in Ethiopia, Evidence from Demographic and Health Survey 2016

PLOS ONE

Dear Mr Muluneh,

Thank you for submitting your manuscript to PLOS ONE. After careful consideration, we feel that it has merit but does not fully meet PLOS ONE’s publication criteria as it currently stands. Therefore, we invite you to submit a revised version of the manuscript that addresses the points raised during the review process.

Please revise to address the final comments and questions from the reviewers. In addition, complete a serious edit to address any outstanding issues (see example from Review 1).

We would appreciate receiving your revised manuscript by Jun 19 2020 11:59PM. To enhance the reproducibility of your results, we recommend that if applicable you deposit your laboratory protocols in protocols.io, where a protocol can be assigned its own identifier (DOI) such that it can be cited independently in the future. For instructions see: http://journals.plos.org/plosone/s/submission-guidelines#loc-laboratory-protocols

We look forward to receiving your revised manuscript.

Kind regards,

Bruce A Larson

Academic Editor

PLOS ONE

Reviewers' comments:

Reviewer's Responses to Questions

**Comments to the Author**

1. If the authors have adequately addressed your comments raised in a previous round of review and you feel that this manuscript is now acceptable for publication, you may indicate that here to bypass the “Comments to the Author” section, enter your conflict of interest statement in the “Confidential to Editor” section, and submit your "Accept" recommendation.

Reviewer #1: (No Response)

Reviewer #2: (No Response)

2. Is the manuscript technically sound, and do the data support the conclusions?

Reviewer #1: Yes

Reviewer #2: Yes

3. Has the statistical analysis been performed appropriately and rigorously? 

Reviewer #1: Yes

Reviewer #2: I Don't Know

4. Have the authors made all data underlying the findings in their manuscript fully available?

Reviewer #1: Yes

Reviewer #2: Yes

5. Is the manuscript presented in an intelligible fashion and written in standard English?

Reviewer #1: Yes

Reviewer #2: Yes

6. Review Comments to the Author

Reviewer #1: Most of my previous comments have been addressed.

However, I feel the manuscript can still benefit from further language editing.

For example, I copied the following section from the Background of the Abstract

"Maternal continuums of care are vital for reducing the mortality of mothers and

neonates. The proportion of dropouts from the care maintained as a rising phenomenon, while

studies on the risk factors associated with defaulters are markedly limited."

The second sentence in the excerpt above is not clear

There are still a couple of sentences such as these throughout which affect smooth readability of the manuscript

Reviewer #2: Thank you for your revisions. Just a couple comments to further improve the clarity of the paper, where I feel they were not addressed.

For instance, the authors say a legend exists for Table 4 as requested, which explain uncommon variables like media exposure (it is not self-explanatory like for instance, age). I am not seeing the legend in revised version. This can actually be one sentence under Table 1 and does not need to repeated for other tables.

Line 170-171 still needs to be fixed like how the authors addressed it in the comment to reviewer. It is still unclear in the manuscript. This is the comment to the reviewer "Among women who booked for ANC only 308 (6.56%) used the complete continuum of care. Among women who had a recommended ANC (four and above) 11.84% used the complete continuum of care." This is much clearer than the way it is currently put in the manuscript

Line 219-221 it talks about proportions for both Debremarkos and Cambodia - these two places share the same statistics?

7. PLOS authors have the option to publish the peer review history of their article (what does this mean?). If published, this will include your full peer review and any attached files.

Reviewer #1: No

Reviewer #2: No

---

## [Author Response · Author response to Decision Letter 1]

22 May 2020

Authors’ response of reviewer 1 

REVIEWER 1 COMMENT AUTHORs RESPONSE

1. Reviewer #1: Most of my previous comments have been addressed.

However, I feel the manuscript can still benefit from further language editing.

For example, I copied the following section from the Background of the Abstract

"Maternal continuums of care are vital for reducing the mortality of mothers and neonates. The proportion of dropouts from the care maintained as a rising phenomenon, while studies on the risk factors associated with defaulters are markedly limited."

The second sentence in the excerpt above is not clear

There are still a couple of sentences such as these throughout which affect smooth readability of the manuscript Thank you for your suggestion!

• We try to re-read and edit some grammatical errors of the whole document. 

• The second sentence we are intended to show dropout from maternal continuum of cares remains high but researches related with dropout and associated factors were limited”

o We have modified as “Maternal continuums of care were vital to reducing maternal and neonatal mortalities. While the dropout rate remains high and limited studies were found on risk factors associated with a high dropout rate of the maternal continuum of care” see the background of the Abstract section. 

Reviewer 2 

REVIEWER COMMENT AUTHORs RESPONSE

1. For instance, the authors say a legend exists for Table 4 as requested, which explain uncommon variables like media exposure (it is not self-explanatory like for instance, age). I am not seeing the legend in revised version. This can actually be one sentence under Table 1 and does not need to repeat for other tables.

 Media exposure: Media exposure was calculated from the internet use, TV watching, radio listening, reading newspapers and those who score above the median were considered as having media exposure and the rest considered as having no media exposure. See line 155-156 below table 1.

2. Line 170-171 still needs to be fixed like how the authors addressed it in the comment to reviewer. It is still unclear in the manuscript. This is the comment to the reviewer "Among women who booked for ANC only 308 (6.56%) used the complete continuum of care. Among women who had a recommended ANC (four and above) 11.84% used the complete continuum of care." This is much clearer than the way it is currently put in the manuscript Thank you!

• We accepted and incorporate the reviewers comment. See line 157-159

3. Line 219-221 it talks about proportions for both Debremarkos and Cambodia - these two places share the same statistics? Thank you very much.

We acknowledge and the magnitude of dropout from continuum of care in Cambodia was missed. About 90 % had at least one ANC, 60% of them 4 and above, 74 % Skill birth attendance and 71% had at least one postnatal checkup. We incorporate it from the main manuscript. See line 203-205 

Answers for Editors’ comment/question

Editor COMMENT AUTHORs RESPONSE

1. Please amend the title either on the online submission form or in your manuscript so that they are identical

 We amended at as the online submitted one and see line 1-3 of the manuscript 

4. You have indicated that data is available from http://www.dhsprogram.com. Before we proceed with your manuscript, please address the following issues in your Data Availability Statement:

a) Please provide a direct link to the database(s) used in your study, or name the specific database(s) used.

b) Please remove "The authors prepared the data that was used for preparation of this manuscript can be shared if required."

Your manuscript has been returned to your account. Please log on to PLOS Editorial Manager at https://www.editorialmanager.com/pone/ to access your manuscript.

Thank you!

• We remove the sentence” The authors prepared the data that was used for preparation of this manuscript can be shared if required”

• We write the name of the specific data base “We used the Ethiopian Demographic and Health Survey 2016 women data” but specific link is not available for this data base. Anyone could register as an authorized user from the major Demographic and Health Survey website http://www.dhsprogram.com and could request specific data set of any countries. See line 279-281 of the manuscript.

5. Please confirm whether the following Data Availability Statement looks acceptable.

"The data used for the preparation of this manuscript were taken from the "Ethiopian Demographic and Health Survey 2016". Users can access the data at http://www.dhsprogram.com."

 Thank you,

Accepted and corrected accordingly. See line 279-280

---

## [Editor Report · Decision Letter 2]

2 Jun 2020

High dropout rate from maternity continuum of care after antenatal care booking and its associated factors among reproductive age women in Ethiopia, Evidence from Demographic and Health Survey 2016

PONE-D-20-01331R2

Dear Dr. Muluneh,

We are pleased to inform you that your manuscript has been judged scientifically suitable for publication and will be formally accepted for publication once it complies with all outstanding technical requirements.

With kind regards,

Bruce A Larson

Academic Editor

PLOS ONE
---

## [Editor Report · Acceptance letter]

4 Jun 2020

PONE-D-20-01331R2 

High dropout rate from maternity continuum of care after antenatal care booking and its associated factors among reproductive age women in Ethiopia, Evidence from Demographic and Health Survey 2016 

Dear Dr. Muluneh:

I'm pleased to inform you that your manuscript has been deemed suitable for publication in PLOS ONE. Congratulations! Your manuscript is now with our production department. 

Kind regards, 

on behalf of

Dr. Bruce A Larson 

Academic Editor

PLOS ONE